# *Oropouche orthobunyavirus* in Urban Mosquitoes: Vector Competence, Coinfection, and Immune System Activation in *Aedes aegypti*

**DOI:** 10.3390/v17040492

**Published:** 2025-03-28

**Authors:** Silvana F. de Mendonça, Lívia V. R. Baldon, Yaovi M. H. Todjro, Bruno A. Marçal, Maria E. C. Rodrigues, Rafaela L. Moreira, Ellen C. Santos, Marcele N. Rocha, Isaque J. da S. de Faria, Bianca D. M. Silva, Thiago N. Pereira, Amanda C. de Freitas, Myrian M. Duarte, Felipe C. de M. Iani, Natália R. Guimarães, Talita E. R. Adelino, Marta Giovanetti, Luiz C. J. Alcantara, Álvaro G. A. Ferreira, Luciano A. Moreira

**Affiliations:** 1Mosquitos Vetores, Endossimbiontes e Interação Patógeno-Vetor, Instituto René Rachou-Fiocruz, Belo Horizonte 30190-002, Brazil; 2Departamento de Bioquímica e Imunologia, Instituto de Ciências Biológicas, Universidade Federal de Minas Gerais, 6627-Pampulha, Belo Horizonte 31270-901, Brazil; 3Laboratório de Cultivo Celular e Isolamento Viral, Fundação Ezequiel Dias, Rua Conde Pereira Carneiro, 80 Bairro Gameleira-Belo Horizonte, Belo Horizonte 30510-010, Brazil; 4Central Public Health Laboratory of the State of Minas Gerais, Ezequiel Dias Foundation, Rua Conde Pereira Carneiro, 80 Bairro Gameleira-Belo Horizonte, Belo Horizonte 30510-010, Brazil; 5Department of Sciences and Technologies for Sustainable Development and One Health, University of Campus Bio-Medico, 00128 Rome, Italy

**Keywords:** *Oropouche orthobunyavirus*, *Aedes aegypti*, *Culex quinquefasciatus*, vector competence, immune system, arbovirus

## Abstract

*Oropouche orthobunyavirus* (OROV) is an emerging public health concern due to its expanding geographic range and increasing case numbers. In Brazil, 13,785 cases were confirmed in 2024, with an additional 3680 reported by January 2025, according to the Ministry of Health. Initially restricted to the Amazon region, OROV has recently been detected in new areas, highlighting the need for enhanced surveillance and vector control strategies. While *Culicoides paraensis* is the primary vector, the potential role of other species in transmitting the currently circulating OROV strain in Brazil remains unclear. Here, we experimentally assessed the infectivity and dissemination of a recently isolated *Oropouche orthobunyavirus* (OROV) strain in two widespread mosquito species, *Aedes aegypti* and *Culex quinquefasciatus*, collected from diverse regions of Brazil. Our results demonstrated that both mosquito species were refractory to oral infection, suggesting that natural transmission through these vectors is unlikely. However, in artificial systemic infection, *Ae. aegypti* showed viral replication and immune system activation, indicating its potential to support OROV replication under specific conditions. Additionally, to assess the potential impact of coinfection, we investigated whether *Chikungunya virus* (CHIKV), an arbovirus that naturally infects *Ae. aegypti*, could facilitate OROV infection dynamics in this mosquito species. Our results suggest that coinfection does not promote OROV oral infection. Furthermore, we examined whether OROV systemic infection induced an immune response in *Ae aegypti.* We analyzed the major immune response pathways—RNAi, Toll, IMD, and JAK-STAT—and observed that the RNAi pathway was the most strongly activated in response to OROV infection in *Ae. aegypti*. These findings highlight the importance of ongoing surveillance and further studies on OROV evolution, vector adaptation, and transmission dynamics, particularly in urban settings where vector populations and viral interactions may facilitate new epidemiological scenarios.

## 1. Introduction

*Oropouche orthobunyavirus* (OROV, Peribunyaviridae) is an arbovirus responsible for an acute, self-limiting febrile illness known as Oropouche Fever, primarily affecting the Amazon region. The main symptoms include fever, headache, body pain, arthralgia, myalgia, and photophobia. In some cases, complications such as hemorrhagic manifestations, meningitis, and neurological impairment have been reported [1,2]. Recent studies have raised concerns about a potential association between OROV infection and congenital malformations, including microcephaly in newborns in Brazil. Although the vertical transmission of OROV has not been conclusively confirmed, the detection of OROV IgM in six newborns with microcephaly, along with OROV RNA and antigen in multiple tissues in a deceased infant, suggests a possible role in fetal development complications [3]. Despite these emerging concerns, there are currently no vaccines or specific antiviral treatments for OROV. Clinical management remains limited to symptomatic treatment, highlighting the need for further research to clarify the risks of maternal–fetal transmission and to develop targeted therapeutic and preventive measures [4].

The virus circulates in both urban and sylvatic cycles, though it primarily persists in sylvatic environments. In the sylvatic cycle, potential vector species include mosquitoes such as *Aedes serratus*, *Coquillettidia venezuelensis*, and *Culex quinquefasciatus* [5]. However, greater concerns arise when urban cycles emerge as human transmission becomes more frequent. In the urban cycle, the primary known vector is *Culicoides paraensis*, a highly anthropophilic insect widely distributed across the Americas. Between 2023 and 2024, there was a significant increase in OROV infections, with reported fatalities associated with the disease and the detection of viral circulation beyond the Amazon region [6,7].

Given the significant increase in cases of Oropouche Fever between 2023 and 2024, including the geographic expansion of the virus and the occurrence of severe complications, it is necessary to investigate the transmission dynamics and potential vectors involved in this recent epidemiological scenario. The surveillance of circulating viruses and the identification of new potential vectors are essential to understanding the spread of OROV and anticipating future outbreaks. In this study, we evaluated the vector competence of mosquitoes collected in 2024 for OROV, as well as the susceptibility of the *Ae. aegypti* (White Eye) strain to the virus. Additionally, we explored the possibility of coinfection between OROV and *Chikungunya virus* (CHIKV) and assessed the activation of the immune system in *Ae. aegypti* during OROV systemic infection. Our findings have broadened the understanding of the virus’s epidemiology, its interaction with other pathogens, and the vector’s response mechanisms, providing insights into more effective control and prevention strategies.

## 2. Materials and Methods

### 2.1. Mosquito Lineages and Feeding

In this study, we worked with two mosquito species: *Ae. aegypti* and *Cx. quinquefasciatus*. For *Ae. aegypti*, we used field populations (F2) from the Brazilian cities of Ouro Preto, Minas Gerais; Joinville, Santa Catarina; and Londrina, Paraná and laboratory strains from Bangkok, Thailand, and Atlanta, USA. For *Cx. quinquefasciatus*, we used populations from the states of Pernambuco and Amazonas, Brazil, both of which had been maintained in the laboratory for over 50 generations. The experiments were conducted in an NB-2 laboratory, in accordance with Extrato de Parecer Técnico nº 6802/2020, published in the Diário Oficial da União, Section 1, on 11 February 2020 [8]. All mosquitoes were kept in insectaries at a temperature of 28 ± 2 °C, relative humidity of 70 ± 10%, and a 12:12 h light/dark photoperiod. For *Cx. quinquefasciatus*, the photoperiod was inverted. Eggs were placed in plastic basins containing 2 L of filtered water. After hatching, *Ae. aegypti* larvae were fed daily with TetraMin^®^ Tropical Tablets (TETRA, Melle, Germany) while *Cx. quinquefasciatus* larvae were fed with floating Goldfish Colour feed (Alcon, Santa Catarina, Brazil), maintaining a density of approximately 200 larvae per basin. After emergence, adults were kept in 30 × 30 × 30 cm BugDorm-1 insect cages (Megaview Science, Taichung, Taiwan) with ad libitum access to 10% sucrose solution.

### 2.2. Production of Viral Stock and Titration

The viral isolate 02.24OROV was used in this study. It was obtained in the first half of 2024 from the blood of a patient in the city of Botuverá, in the state of Santa Catarina (SC), Brazil, with GenBank accession numbers PQ247721, PQ168305, and PQ168436. RT-qPCR was performed on this sample, and the cycle threshold (Ct) value obtained was 16.13. The 02.24OROV isolate was kindly provided by Dr. Myrian Morato Duarte from the Laboratory of Cell Culture and Viral Isolation at the Ezequiel Dias Foundation (Funed), Minas Gerais, Brazil.

For CHIKV, we used the CHIKV-ECSA strain isolated from a human patient in Niterói, Rio de Janeiro state, Brazil, in 2018 (GenBank number—MK244647) [9]. For DENV, we used DENV-1/H. sapiens/Brazil/Contagem/MG/BRMV09/2015), originally isolated from human blood in Contagem, Minas Gerais, Brazil, in 2015 [10].

Vero cells were maintained in high-glucose Dulbecco’s Modified Eagle Medium (DMEM-High, Sigma-Aldrich, St. Louis, MO, USA) supplemented with 5% fetal bovine serum (FBS, Sigma-Aldrich, St. Louis, MO, USA) and 1% penicillin and streptomycin. C6/36 cells were maintained in Leibovitz’s L15 Medium (L15, Sigma-Aldrich, St. Louis, MO, USA) enriched with 10% fetal bovine serum (FBS, Sigma-Aldrich, St. Louis, MO, USA) and 1× Antibiotic Antimycotic (Sigma-Aldrich, St. Louis, MO, USA). One day before viral adsorption, cells were seeded in 25 cm^2^ cell culture flasks (Sarstedt, Nümbrecht, Germany) to reach 70% confluence. The cells were infected at a multiplicity of infection (MOI) of 0.01. The adsorption process was performed using the virus and low-glucose Dulbecco’s Modified Eagle Medium (DMEM-Low, Sigma-Aldrich, St. Louis, MO, USA) for 1 h, with the flasks being agitated every 10 min. At the end of the adsorption period, DMEM-High or L15 was added to a final volume of 5 mL per flask, and the cultures were incubated for 3 to 4 days, for CHIKV and OROV, in an incubator (Thermo Fisher Scientific, Waltham, MA, USA) at 37 °C with 5% CO_2_, and 6 days for DENV in an incubator BOD at 28 °C. After this period, the supernatant was collected, and the cells were lysed by repeated freeze–thaw cycles to release viral particles, which were then mixed with the supernatant. The resulting supernatant was clarified by centrifugation, and the viral aliquots were stored at −80 °C until further use.

The titration of viral stocks was performed in Vero cells using the plaque-forming unit (pfu) quantification method. A six-well cell culture plate was used for this process. Adsorption occurred over 1 h, with the plates incubated at 37 °C and 5% CO_2_, being agitated every 10 min. After this period, a 2% carboxymethylcellulose (CMC) (Synth, São Paulo, Brazil) overlay diluted in DMEM-High supplemented with 2% fetal bovine serum (FBS) and 1% penicillin/streptomycin was added to each well. The plates were incubated for 5 days and then fixed with 3.7% formaldehyde. After fixation, the plates were washed to remove the CMC and stained with crystal violet solution (70% water, 30% ethanol, and 0.25% crystal violet) for about 1 h. The plates were then washed again and dried and the lysis plaques were counted. Titration was calculated based on the number of lysis plaques observed in the well containing between 30 and 300 plaques. This value was multiplied by the inverse of the dilution factor and the correction factor (1000/500 = 2). The final titer was expressed in plaque-forming units per milliliter (pfu/mL), totaling 1.5 × 10^5^ pfu/mL for OROV, 7 × 10^6^ pfu/mL for CHIKV, and 1 × 10^7^ pfu/mL for DENV.

### 2.3. Infection of Mosquitoes with OROV

For oral infection via membrane feeding, adult females, approximately 5 to 7 days old, were starved for 24 h before being fed a mixture containing blood and the supernatant of virus-infected cell cultures (at a ratio of 1 mL of virus to 1 mL of blood). A glass artificial feeder system coated with a pig intestine membrane was used. After the feeding period, which lasted 1 h, the females were anesthetized with CO_2_, and fully engorged mosquitoes were separated and housed in cages until they reached 8 days post infection (dpi), at which point they were collected for molecular biology analysis. For systemic infection, each female was intrathoracically injected with 70 nL of virus using the Nanoject III nanoinjector (Drummond Scientific, Broomall, PA, USA). Mock-injected mosquitoes received 70 nL of uninfected cell culture supernatant. After 8 dpi, they were collected for analysis.

### 2.4. RNA Extraction and RT-qPCR

The RNA from whole mosquito samples of both tested species was individually extracted using Trizol^®^ reagent (Thermo Fisher Scientific, Waltham, MA, USA), following the manufacturer’s protocol, as previously performed by [11]. Briefly, the tissue was homogenized in a microtube containing 2 glass beads and 200 μL of Trizol, using the Mini-Beadbeater-96 (Biospec Products, Bartlesville, OK, USA) for 1 min and 30 s. Then, 40 μL of chloroform was added, homogenized using a vortex, and incubated at room temperature for 5 min. Phase separation was achieved by centrifugation at 12,000 rcf for 15 min at 4 °C. The supernatant (upper transparent phase), containing RNA, was collected and transferred to a new tube containing 4 μL of glycogen (5 mg/mL) (Thermo Fisher Scientific, Waltham, MA, USA). RNA precipitation was performed by adding 100 μL of isopropanol (Sigma-Aldrich/Merck, St. Louis, MO, USA) and incubating overnight, followed by centrifugation at 12,000 rcf for 10 min at 4 °C. The supernatant was then discarded, and the pellet was washed with 75% ethanol, resuspended in 20 μL of RNase- and DNase-free water (Ultra Pure Distilled Water— Thermo Fisher Scientific, Waltham, MA, USA), and stored at −70 °C.

For the RNA samples from *Cx. Quinquefasciatus* mosquitoes and for the samples from the gene expression experiment, cDNA synthesis was performed. The reverse transcription protocol was preceded by a Dnase treatment step, carried out according to the manufacturer’s instructions for the M-MLV RT enzyme (Promega, Madison, WI, USA). For cDNA synthesis, the Dnase-treated RNA sample was incubated with random primers (Promega, Madison, WI, USA) and water at 70 °C for 5 min, then immediately cooled on ice for 2 min. Meanwhile, a reaction mix was prepared containing 4 μL of 5× Reaction Buffer from the M-MLV enzyme (Tris-HCl 250 mM pH 8.3, KCl 375 mM, MgCl_2_ 15 mM, and DTT 50 mM), 2.5 μL of 10 mM dNTPs, 0.5 μL of M-MLV RT enzyme, and 1 μL of RNase-free water. This mix was added to the first reaction component, making a final volume of 20 μL, and incubated at 37 °C for 1 h, followed by 70 °C for 10 min. For negative controls, reverse transcriptase was omitted. Finally, the cDNA sample was diluted at a 1:10 ratio.

All samples were subjected to either relative or absolute quantification using real-time PCR. For *Ae. aegypti* samples, except for the gene expression experiment, the GoTaq^®^ Probe 1-Step RT-qPCR System reagent (Promega, Madison, WI, USA) was used, following the manufacturer’s protocol, along with HardShell^®^ 96-Well PCR plates (Bio-Rad Laboratories, Hercules, CA, USA). The real-time PCR was performed on a QuantStudio 12K Real-Time PCR System (Applied Biosystems, Foster City, CA, USA) under the following cycling conditions: pre-incubation at 45 °C for 5 min; reverse transcriptase inactivation and GoTaq DNA polymerase activation at 95 °C for 2 min; and 35 cycles of PCR amplification at 95 °C for 3 s, then 60 °C for 30 s. Each reaction had a total volume of 10 µL: 5 µL GoTaq^®^ Probe qPCR Master Mix with dUTP (2X), 0.2 µL GoScript™ RT Mix for 1-Step RT-qPCR, 0.5 µL of 10 µM OROV or CHIKV primers (Forward + Reverse), 0.5 µL of 10 µM *Ae. aegypti* primers (Forward + Reverse), 0.1 µL of 10 µM OROV or CHIKV probe, 0.1 µL of 10 µM *Ae. aegypti* probe, 1.1 µL of nuclease-free water, and approximately 40 ng of total extracted RNA. For *Cx. quinquefasciatus*, and for the samples from the gene expression experiment, the SYBR Green PCR Master Mix kit (Applied Biosystems—Life Technologies, Foster City, CA, USA) was used. Each reaction had a final volume of 10 µL. The PCR cycling conditions were as follows: initial step at 95 °C for 20 s; PCR amplification for 40 cycles at 95 °C for 15 s, followed by 60 °C for 60 s; and a melting curve analysis from 60 °C to 95 °C, with a final step of 15 s at each temperature point. When the GoTaq^®^ Probe 1-Step RT-qPCR System reagent was used, samples were tested in duplicates, whereas when the SYBR Green PCR Master Mix kit was used, samples were tested in triplicates. Gene expression was quantified using the 2^−^∆Ct method. The cycle threshold (Ct) values for the target genes (OROV and CHIKV) were normalized to the Ct values of the internal reference gene (RPS17, RPL32, or 18S) within the same sample. Viral RNA levels were calculated relative to the endogenous control gene RPS17 or RPL32 for *Ae. aegypti* and 18S for *Cx. quinquefasciatus*. The primers used for RNA quantification are listed in Table 1.

### 2.5. Statistical Analysis

Statistical analyses were performed with R software version 4.3.3 (29 February 2024). The Wilcoxon–Mann–Whitney test (Wilcoxon.test) was used for comparing viral load between only two groups while the Kruskal–Wallis test (Kruskal.test) was performed to compare viral load across multiple samples. Both tests assumed a *p*-value < 0.05 for consideration of statistically significant differences.

### 2.6. Ethics Committee

The human blood used in this study was provided by the blood bank of the Hemominas Foundation, in accordance with the cooperation agreement with the René Rachou Institute, Fiocruz/MG, under the registration OF.GPO/CCO- Nr 224/16.

## 3. Results

### 3.1. Ae. aegypti and Cx. quinquefasciatus Are Refractory to OROV Infection When Fed Orally

To assess the vector competence of *Ae. aegypti* and *Cx. quinquefasciatus*, females aged approximately 5 to 7 days were fasted for about 24 h and then subjected to a blood meal in an artificial feeder containing blood and virus in a 1:1 ratio (one part virus to one part blood). After feeding, the engorged females were separated and analyzed by RT-qPCR at 8 days post infection. No infection was detected in any of the *Ae. aegypti* populations analyzed (Figure 1A), indicating that both the laboratory population (BKK) and field populations from the Brazilian States of Minas Gerais (MG), Santa Catarina (SC), and Paraná (PR) were refractory to oral infection. Similar results (Figure 1B) were observed in *Cx. quinquefasciatus* for populations from the states of Pernambuco (PE) and Amazonas (AM).

To investigate whether *Ae. aegypti* mutant lines with midgut homeostasis deficiency exhibited altered susceptibility to viral infection, we tested the White Eye (WE) lineage of *Ae. aegypti.* This lineage is characterized by a lack of xanthurenic acid (XA) production due to a deficiency in kynurenine hydroxylase, leading to kynurenic acid accumulation instead [12]. As a result, WE mosquitoes experience increased midgut epithelial cell death following blood feeding, underscoring its role as an antioxidant that mitigates heme- and iron-induced oxidative stress. Here, the WE lineage was used as a model of potential susceptibility to infection due to its XA deficiency while mosquitoes from the BKK lineage served as controls. Additionally, we used dengue virus (DENV) as a comparative infection model. As shown in Figure 2, no OROV infection was detected in either mosquito lineage, in contrast to the successful DENV infection. However, no significant difference in infection rates was observed between BKK and WE populations across replicates (*p* = 0.4931).

### 3.2. Coinfection with CHIKV Does Not Alter OROV Infection

After confirming that *Ae. aegypti* mosquitoes from different populations were refractory to *Oropouche orthobunyavirus* (OROV) infection through artificial feeding, we sought to investigate whether coinfection with another arbovirus known to be transmitted by *Aedes* mosquitoes could facilitate OROV infection. To assess this, *Ae. aegypti* BKK females were exposed to an infectious blood meal in three distinct experimental groups: (i) OROV-only, receiving blood containing OROV; (ii) CHIKV-only, receiving blood containing Chikungunya virus (CHIKV); and (iii) coinfection, receiving blood containing both OROV and CHIKV (Figure 3). Our results revealed that OROV infection was not detected in either the OROV-only or coinfected groups, indicating that CHIKV coinfection does not promote OROV susceptibility. In contrast, CHIKV infection was observed in both relevant groups: an 84% infection rate in the CHIKV-only group and 67% in the coinfected group (Figure 3).

### 3.3. Ae. aegypti and Cx. quinquefasciatus Become Infected with OROV Through Systemic Infection

Previous studies conducted by our research group have demonstrated that when systemically injected into the mosquito body cavity, both *Ae. aegypti* and *Cx. quinquefasciatus* are capable of becoming infected, replicating, and transmitting OROV to a vertebrate host [13]. In this study, we further evaluated the systemic infection dynamics of the currently circulating OROV strain in Brazil in *Ae. aegypti* using the intrathoracic injection method. As shown in Figure 4, all tested *Ae. aegypti* populations (BKK, MG, SC, PR) exhibited 100% infection rates with high viral loads, with no significant differences between populations (*p* > 0.05). In contrast, for *Cx. quinquefasciatus*, both infection rates and viral loads were low in the two populations analyzed, indicating a reduced susceptibility to OROV systemic infection compared to *Ae. aegypti.*

### 3.4. The Immune System of Ae. aegypti Plays a Role in OROV Infection

Given that *Ae. aegypti* mosquitoes can be systemically infected with OROV and support viral replication, we investigated whether OROV infection induces an immune response in the mosquito host. Following systemic infection, we quantified the expression of key immune-related genes representing the major antiviral immune pathways in mosquitoes. Specifically, we examined Argonaute 2 (Ago2) from the RNA interference (RNAi) pathway, Relish2 and Caspar from the IMD pathway, Cactus from the Toll pathway, and Suppressor of Cytokine Signaling (SOCS) from the JAK-STAT pathway. Gene expression levels were analyzed at 4 and 8 days post infection (dpi) to evaluate potential immune activation over time in response to OROV infection.

Before investigating the putative activation and induction of immune pathways, we first assessed whether the mosquitoes used in our assays became infected with OROV following systemic injection. As shown in Figure 5A, our results confirm the successful systemic infection of *Ae. aegypti* mosquitoes after OROV inoculation. Having established that OROV infection occurs in these mosquitoes, we next sought to determine whether this infection triggers an immune response. Given the well-characterized role of the RNA interference (RNAi) pathway in antiviral defense, we examined Ago2 expression as a marker of RNAi activation. As shown in Figure 5B, we observed a significant increase in Ago2 expression at 4 dpi in OROV-infected mosquitoes (*p* = 0.02). Furthermore, at 8 dpi, Ago2 expression was even more pronounced, with a highly significant increase (*p* = 1.42 × 10^−10^), suggesting a strong activation of the RNA interference (RNAi) pathway in response to OROV infection. We next analyzed the IMD pathway, which is known to regulate bacterial and viral immune responses and potential involved in antiviral immune responses. Next, we analyzed the IMD pathway, which is known to regulate immune responses against both bacterial and viral infections and may play a role in antiviral immunity. The expression of Relish2, a key signaling molecule in the IMD pathway, was significantly higher in OROV-infected mosquitoes at 4 dpi (*p* = 0.02) and further increased at 8 dpi (*p* = 1.31 × 10^−4^). These results indicate that the IMD pathway is also activated in response to systemic OROV infection.

In contrast, when we analyzed Cactus expression, a negative regulator of the Toll pathway, we found that it remained unchanged at 4 dpi (*p* = 0.54) but was significantly upregulated at 8 dpi (*p* = 0.01) in OROV-infected mosquitoes (Figure 6B). This suggests a delayed activation of the Toll pathway in response to viral infection, potentially playing a role in the mosquito’s immune response over time. Similarly, for the JAK-STAT pathway, we observed no significant change in SOCS expression at 4 dpi (*p* = 0.18). However, by 8 dpi, SOCS was significantly downregulated (*p* = 1.88 × 10^−5^) in OROV-infected mosquitoes compared to controls (Figure 6C). Since SOCS is a negative regulator of JAK-STAT signaling, its suppression suggests that the JAK-STAT pathway may be activated in response to OROV infection, potentially contributing to antiviral immunity or immune modulation in *Ae. aegypti.*

Together, these findings suggest that while OROV infection is established in *Ae. aegypti*, the activation of antiviral immune pathways may be heterogeneous, with potential differences in the RNAi and IMD responses among individuals. Further studies are needed to determine the functional relevance of these immune responses in restricting OROV replication.

## 4. Discussion

The increasing number of OROV infection cases recorded in 2024 across various regions of Brazil and South American countries highlights the need to intensify surveillance and control strategies for emerging and reemerging arboviruses. This study analyzed the vector competence of two cosmopolitan and highly anthropophilic species, *Ae. aegypti* and *Cx. quinquefasciatus*, for the OROV strain recently isolated in Brazil. Although *C. paraensis* remains the primary known vector to date, the detection of the virus circulating in regions where it was previously absent, along with the potential for viruses to adapt to new transmission cycles, justifies investigating other species as potential vectors.

Our results demonstrate that both *Ae. aegypti* and *Cx. quinquefasciatus* are refractory to oral OROV infection, indicating that natural transmission via these mosquito species is unlikely. However, systemic infection experiments revealed that *Ae. aegypti* supports high levels of OROV replication, while *Cx. quinquefasciatus* exhibited low infection rates and viral loads following intrathoracic injection. This difference in susceptibility suggests that *Ae. aegypti* may possess intrinsic cellular factors that facilitate OROV replication if the midgut infection barrier is bypassed. However, its role in natural transmission remains uncertain.

Our results suggest that the midgut infection barrier plays a critical role in preventing OROV replication in *Ae. aegypti* and *Cx. quinquefasciatus* as neither species became infected following oral exposure. This finding aligns with previous studies demonstrating that the midgut acts as a key determinant of vector competence, restricting viral entry and dissemination in various mosquito–arbovirus systems. Rather than being dependent on specific environmental or physiological conditions, our data support the idea that the midgut barrier itself may prevent OROV from establishing infection in these mosquitoes. Additionally, the viral dose used in our oral infections may have influenced these results as higher viral titers have been shown to increase infection rates in some mosquito–virus combinations. While we did not assess the effect of increasing OROV titers in this study, future experiments testing a range of viral doses could help clarify whether the observed refractoriness is an absolute midgut infection barrier or a dose-dependent effect. These findings underscore the importance of considering both intrinsic barriers and viral load when assessing the potential of alternative mosquito species to serve as vectors for emergent arboviruses [14,15,16,17].

While our study provides insights into the interactions between OROV and mosquitoes, we acknowledge the importance of comparing these results with *C. paraensis*, the primary vector of OROV. However, as far as we know, there is currently no established method for rearing *C. paraensis* under laboratory conditions, making controlled experimental infections unfeasible. Future studies using field-collected *C. paraensis* may help clarify the natural vector’s response to OROV infection, although logistical challenges remain.

Although *Ae. aegypti* is a competent vector for various arboviruses, such as dengue, Zika, Chikungunya, and Yellow Fever, it has a low capacity to become infected with OROV via the oral route, as we presented in a previously published study [13]. Notably, in our current study, we used different mosquito populations and, more importantly, an OROV recently isolated from a patient in Santa Catarina State, Brazil, in 2024, reflecting the currently circulating genotype in the country.

Despite obtaining similar results, it remains crucial to continuously monitor and study circulating viruses in parallel with mosquito populations collected from the same time and location, as pathogen–vector interactions are dynamic and subject to evolutionary pressures that may lead to adaptations to new transmission cycles [18]. A well-documented example of such an evolutionary shift is that of *Chikungunya virus* (CHIKV), which, after acquiring a single-point mutation in its genome, gained the ability to efficiently infect *Ae. albopictus*, allowing its expansion into regions where *Ae. aegypti* was not predominant [19]. This geographic expansion was accompanied by a surge in CHIKV cases, demonstrating how viral adaptation to new vectors can alter transmission dynamics and pose significant challenges for vector control strategies, underscoring the importance of continuous surveillance and proactive public health measures.

Similarly to *Ae. aegypti*, the potential role of *Cx. quinquefasciatus* in OROV transmission remains uncertain. A previous study detected one OROV genome segment in a pool of *Cx. quinquefasciatus* mosquitoes collected in Mato Grosso, Brazil [20]. However, under laboratory conditions, McGregor et al. (2021) reported low infection rates, with 9.71% at 10 dpi and 19.3% at 14 dpi, while dissemination and transmission rates remained below 2%, reinforcing our findings that *Cx. quinquefasciatus* exhibits low vector competence for OROV. Like *Cx. quinquefasciatus*, *Culex tarsalis*, another widespread *Culex* species, also showed low susceptibility to OROV infection, with an infection rate of only 3.13% in the same study [21]. Additionally, earlier research by Hoch et al. (1987), using hamsters as an infection model, further supported the limited ability of *Cx. quinquefasciatus* to sustain OROV infection [22]. These findings suggest that while *Culex* mosquitoes occasionally acquire OROV, their low infection, dissemination, and transmission rates indicate that they are unlikely to serve as primary vectors for the virus. Differences in mosquito populations, viral strains, and experimental infection methodologies may explain why, in our study, we did not detect any OROV-infected mosquitoes. These findings underscore the importance of ongoing surveillance and experimental validation to fully elucidate the vectorial capacity of different mosquito species and their potential roles in OROV transmission cycles.

Given the complexity of mosquito–virus interactions, another important factor to consider is the potential impact of arbovirus coinfection on vector competence. Coinfection with viruses such as CHIKV has been shown to influence transmission dynamics by modulating the mosquito’s immune response, which can sometimes increase infection rates. A study by Le Coupanec et al. (2017) demonstrated that CHIKV and DENV coinfection facilitated viral replication in *Ae. aegypti* while research by Leggewie et al. (2023) showed that the antiviral immune response of *Ae. aegypti* is activated in both single and arbovirus coinfections [23,24]. Based on these findings, we investigated the impact of CHIKV–OROV coinfection. However, our results showed no OROV infection, suggesting that the effect of coinfection may depend on the mosquito’s inherent vector competence. Since OROV is refractory to infection in the mosquito midgut, coinfection does not appear to facilitate its dissemination, further reinforcing the idea that *Ae. aegypti* is unlikely to play a significant role in OROV transmission.

Since OROV was able to replicate in *Ae. aegypti* following systemic infection, we explored whether viral replication triggered an immune response in the mosquito. We observed a significant upregulation of Argonaute 2 (Ago2), a key component of the RNA interference (RNAi) pathway, suggesting that this pathway plays a role in controlling OROV replication. Our results indicate a delayed induction of Ago2 expression following OROV infection in mosquitoes, with a more pronounced increase at 8 dpi. This suggests that the RNAi pathway, rather than being immediately activated upon viral entry, may play a compensatory role in controlling viral replication at later stages of infection. Unlike Toll and IMD, which are rapidly induced upon bacterial or fungal recognition, the RNAi response relies on the processing of viral double-stranded RNA, which may take time to accumulate to levels sufficient for triggering a robust antiviral response. Additionally, arboviruses that naturally infect mosquitoes have likely evolved strategies to evade or suppress the RNAi pathway, preventing the significant transcriptional upregulation of Ago2. In contrast, OROV is not a natural mosquito-borne virus, and its ability to interfere with RNAi defenses may be limited. This could explain the increased expression of Ago2 at later time points, suggesting that OROV infection may stimulate an antiviral RNAi response when viral replication peaks [25,26,27,28,29]. These findings highlight a potentially important difference in the host–pathogen interactions between endemic arboviruses and those that are experimentally introduced into mosquito models, underscoring the need for further studies to explore the evolutionary dynamics of RNAi evasion in arboviruses.

The IMD pathway, which is also involved in antiviral responses, showed no significant activation, as Caspar expression remained unchanged. However, Relish2, another IMD component, was upregulated, indicating that specific aspects of this pathway may be involved in the immune response to OROV.

A notable finding was the delayed induction of Cactus, a negative regulator of the Toll pathway. While Cactus expression remained unchanged at 4 dpi, it was significantly upregulated at 8 dpi, suggesting that the Toll pathway may have been activated at early stages of infection and subsequently repressed at later stages to restore homeostasis. This delayed regulatory response highlights the dynamic nature of immune signaling, where initial activation may be followed by mechanisms to prevent prolonged immune stimulation and potential cellular damage.

Interestingly, SOCS expression, a negative regulator of the JAK-STAT pathway, was significantly downregulated at 8 dpi, suggesting that this pathway may become less suppressed, leading to increased activity. One possible explanation is that this heightened activation is due to the JAK-STAT pathway’s role in cellular stress response, as OROV replication may induce cellular stress. These findings suggest a complex and dynamic immune response wherein the Toll and JAK-STAT pathways may play distinct roles at different stages of infection, potentially contributing to viral control or immune modulation.

Collectively, our findings provide new insights into the interaction between OROV and *Ae. aegypti*, highlighting the refractory nature of this species in relation to oral infection, the capacity for systemic viral replication, and the immune pathways activated in response to infection. While our data suggest that *Ae. aegypti* is unlikely to serve as a primary vector for OROV, the ability of the virus to replicate in this species under experimental conditions raises important questions regarding the potential for alternative transmission routes, such as venereal or vertical transmission, which should be explored in future studies. Despite our results providing important data on the vector competence of *Ae. aegypti* and *Cx. quinquefasciatus*, some caveats should be mentioned. The main limitation concerns the use of artificial infection, which may not fully reflect the natural conditions of virus transmission. Additionally, the mosquito sample used was limited to certain regions of Brazil, and further studies would be necessary to assess vector competence in different geographical contexts, especially in areas with a higher incidence of OROV.

Given the expanding geographical range of OROV and the increase in human cases, continued vector surveillance is essential to assess the risk of adaptation to new mosquito vectors. Future research should also investigate the molecular mechanisms governing viral replication in refractory species, as well as potential interactions between OROV and other circulating arboviruses in endemic regions.

## Figures and Tables

**Figure 1 viruses-17-00492-f001:**
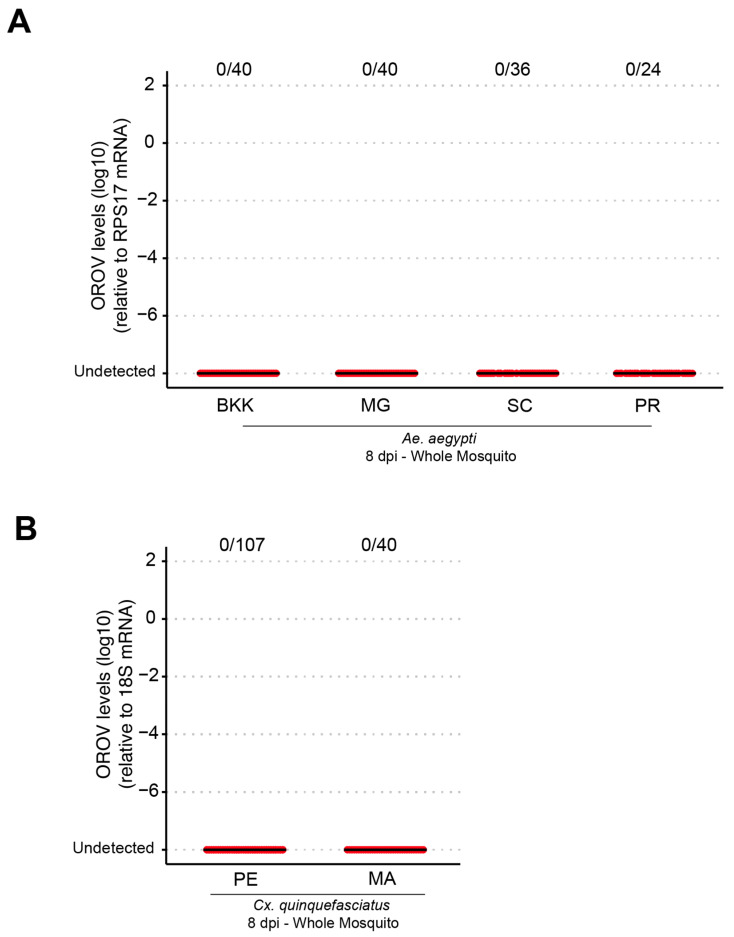
Different populations of *Ae. aegypti* and *Cx. quinquefasciatus* were refractory to OROV infection when fed orally. (**A**) OROV RNA levels in each whole mosquito, at 8 dpi, from *Ae. aegypti* populations from BKK, MG, SC, and PR. (**B**) OROV RNA levels in each whole mosquito, at 8 dpi, from *Cx. quinquefasciatus* populations from PE and MA. The viral titer used was 1.5 × 10^5^ pfu/mL. Each mosquito was analyzed individually, and the total number of infected mosquitoes × total number analyzed for each population is indicated by the numbering at the top of the graph. The data presented correspond to one of the two repetitions of the experiment. The repetition is shown in Appendix A.

**Figure 2 viruses-17-00492-f002:**
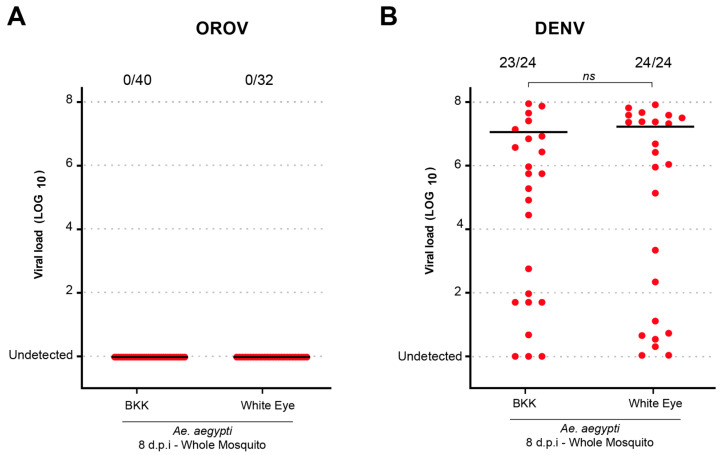
The *Ae. aegypti* White Eye strain does not become infected with OROV when orally fed. (**A**) OROV viral load in *Ae. aegypti* BKK and White Eye in whole mosquitoes at 8 dpi. (**B**) DENV viral load in *Ae. aegypti* BKK and White Eye in whole mosquitoes at 8 dpi. The viral titer used was 1.5 × 10^5^ pfu/mL for OROV and 1 × 10^7^ pfu/mL for DENV. Each mosquito was analyzed individually, and the total number of infected mosquitoes × total number analyzed for each population is indicated by the numbering at the top of the graph. “*ns*” indicates non-significant difference (*p* > 0.05, Mann–Whitney U test). The data presented correspond to one of the two repetitions of the experiment. The repetition is shown in Appendix A.

**Figure 3 viruses-17-00492-f003:**
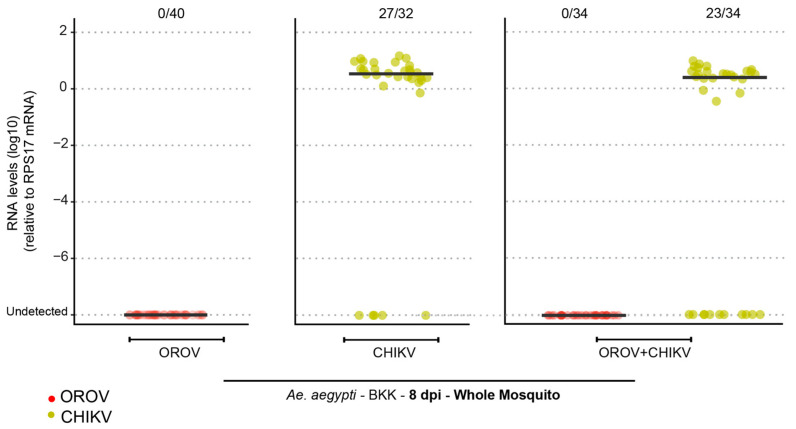
*Ae. aegypti* (BKK) mosquitoes are refractory to OROV infection when coinfected with CHIKV. OROV or CHIKV RNA levels in whole mosquitoes at 8 dpi for the *Ae. aegypti* population from BKK. Red circles indicate mosquitoes analyzed for OROV presence while green circles represent mosquitoes analyzed for CHIKV presence. The viral titer used was 1.5 × 10^5^ pfu/mL for OROV and 7 × 10^6^ pfu/mL for CHIKV. Each mosquito was analyzed individually, and the total number of infected mosquitoes × total number analyzed is indicated by the numbering at the top of the graph. The data presented correspond to one of the two repetitions of the experiment. The repetition is shown in Appendix A.

**Figure 4 viruses-17-00492-f004:**
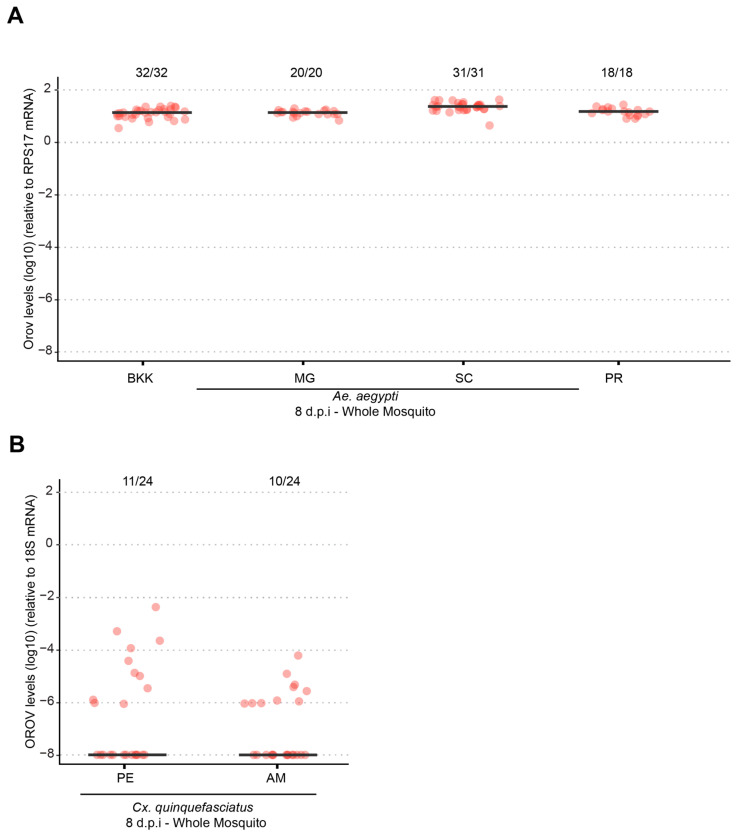
Different populations of *Ae. aegypti* and *Cx. quinquefasciatus* becomes infected with OROV via intrathoracic inoculation. (**A**) OROV RNA levels in whole mosquitoes at 8 dpi for *Ae. aegypti* populations from BKK, MG, SC, and PR. (**B**) OROV RNA levels in whole mosquitoes at 8 dpi for *Cx. quinquefasciatus* populations from PE and MA. The viral titer used was 1.5 × 10^5^ pfu/mL. Each mosquito was analyzed individually, and the total number of infected mosquitoes × total number analyzed for each population is indicated by the numbering at the top of the graph. The data presented correspond to one of the two repetitions of the experiment. The repetition is shown in Appendix A.

**Figure 5 viruses-17-00492-f005:**
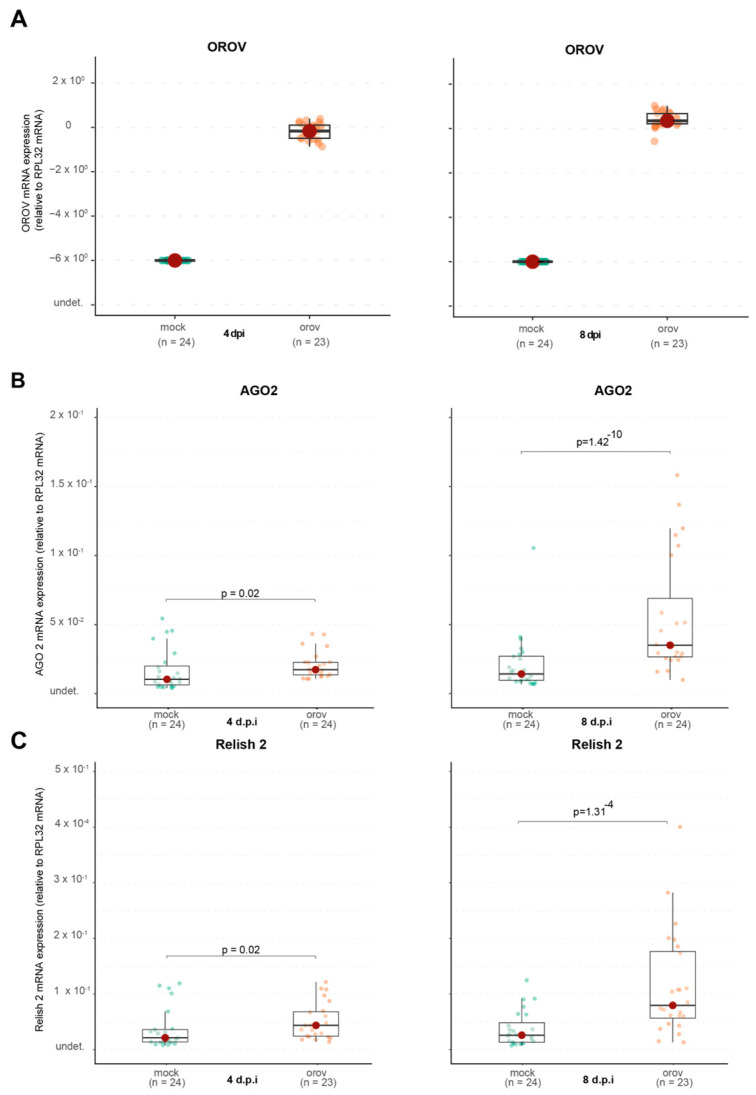
The immune system of *Ae. aegypti* is activated during OROV infection. (**A**) OROV RNA levels in whole mosquitoes. (**B**) AGO2 RNA levels in whole mosquitoes. (**C**) Relish 2 RNA levels in whole mosquitoes. The viral titer used was 1.5 × 10^5^ pfu/mL. Mock-injected mosquitoes received 70 nL of uninfected cell culture supernatant. Each mosquito was analyzed individually at 4 and 8 dpi. Mock-injected mosquitoes are shown in green and OROV-injected mosquitoes are shown in orange. The data presented correspond to one of the two repetitions of the experiment. The repetition is shown in Appendix A. See Figure 6 for continuation.

**Figure 6 viruses-17-00492-f006:**
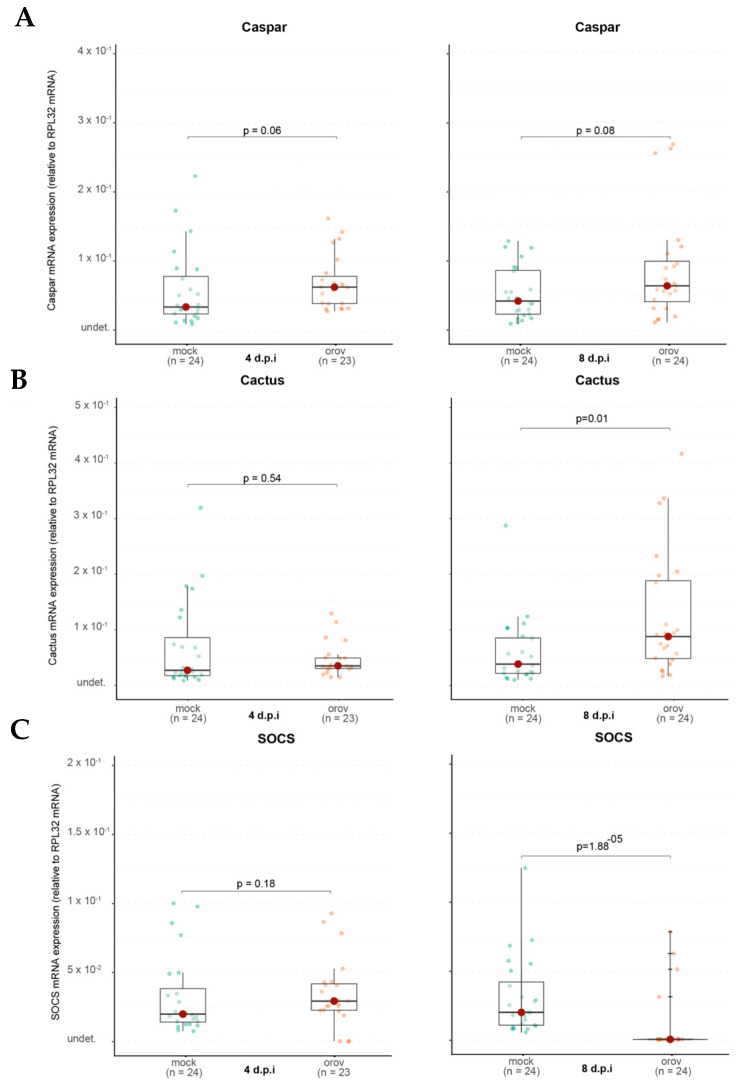
The immune system of *Ae. aegypti* is activated during OROV infection. (**A**) Caspar RNA levels in whole mosquitoes. (**B**) Cactus RNA levels in whole mosquitoes. (**C**) SOCS RNA levels in whole mosquitoes. Each mosquito was analyzed individually at 4 and 8 dpi. Mock-injected mosquitoes received 70 nL of uninfected cell culture supernatant. Mock-injected mosquitoes are shown in green and OROV-injected mosquitoes are shown in orange. The data presented correspond to one of the two repetitions of the experiment. The repetition is shown in Appendix A.

**Table 1 viruses-17-00492-t001:** Primers used for RNA quantification.

Gene	Sequence
*RPS17* *(Ae. aegypti)*	F—5′-TCC GTG GTA TCT CCA TCA AGC T-3′ R—5′-CAC TTC CGG CAC GTA GTT GTC-3′ Probes—5′-HEX/CAG GAG GAG GAA CGT GAG CGC AG/3BHQ2-3′ and 5′-/5FAM/CAG GAG GAG GAA CGT GAG CGC-3′
*RPL32* *(Ae. aegypti)*	F—5′-AGC CGC GTG TTG TAC TCT G-3′ R—5′-ACTTCT TCG TCC GCT TCT TG-3′
*18s* *(Cx. quinquefasciatus)*	F—5′-CGC GGT AAT TCC AGC TCC ACT A-3′ R—5′-GCA TCA AGC GCC ACC ATA TAG G-3′
*OROV* *(Oropouche* *orthobunyavirus)*	F—5′-CAA CGA TGT ACC ACA ACG GAC TAC-3′ R—5′-ACA ACA CCA GCA TTG AGC ACT T-3′ Probe- 5′-/56-FAM/TTG ATC CGG/ZEN/ AGG CAG CAT ATG TGG/3IABkFQ/—3′
*CHIKV* *(Chikungunya virus)*	F—5′-AAG CTY CGC GTC CTT TAC CAA G-3′ R—5′-CCA AAT TGT CCY GGT CTT CCT-3′ Probe—5′-/5HEX/CCA ATG TCY/ZEN TCM GCC TGG ACA CCT TT/3IABkFQ/-3′
*AGO2* *(Argonauta)*	F—5′-TTGTTTGCTTCGTTGCTCTTT-3′ R—5′-ATCTCCTACACCGAACCCACT-3′
*Rel 2* *(Relish 2)*	F—5′-TGAATGTGCTGTTGGGTCAT-3′ R—5′-TTTTTACACATCACCGCCAA-3′
*CASPAR*	F—5′-GAATCCGAGCGAGCCGATGC-3′ R—5′-CGTAGTCCAGCGTTGTGAGGTC-3′
*CACTUS*	F—5′-AGACAGCCGCACCTTCGATTCC-3′ R—5′-CGCTTCGGTAGCCTCGTGGATC-3′
*SOCS*	F—5′-CCGAAATCACTCAAATCCTACC-3′ R—5′-ATCGTCCAGTGGCCTGTATC-3′

## Data Availability

The data presented in this study are openly available at https://doi.org/10.6084/m9.figshare.28505462 (accessed on 21 March 2025).

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
