# Peer review of "Oropouche orthobunyavirus in Urban Mosquitoes: Vector Competence, Coinfection, and Immune System Activation in Aedes aegypti"

_viruses, 2025, doi:10.3390/v17040492_

Round 1
Reviewer 1 Report
Comments and Suggestions for Authors
In this manuscript, Mendonça and colleagues explored the vector competence, impact of co-infection, and immune system activation in Aedes aegypti under Oropouche orthobunyavirus exposure. The manuscript is well-written and contains an important piece of work that will help the literature understand this critical emergent arbovirus.
Howerver, there are some points that should be addressed before its ready for publication:
MINOR
#Topic 2.2
What was the status of the patients from whom the viral strains were retrieved? It could help to understand the pathogenicity level of these isolates.
#Topic 2.3
Why the time point 8dpi was selected?
#Results
Name of viral species accepted by ICTV must be italicized
Why are normalizers different for the experiments? i.e. 18S and RPS17 ... Only because it's a different species?
Authors have to decide whether they use Orov or OROV in the Y-axis.
MAJOR
Since the authors already investigated systemic infection through body injection, I wonder if it wouldn't be more useful to include a known vector to comparatively assess the viral levels and 'systemic susceptibility' according to a natural setting, such as C. paraensis vector. I am mostly curious about the RNA levels...
I am not sure if we expected that Ago2 would be induced during arbovirus infections? It has been seen for another arbovirus? I was surprised by these results, is it corroborated by other similar experiments? In addition, I wonder why not a Knockdown to see its impact on the control of the virus replication?
Again, in the discussion lines 375-381, it would be more informative if compared with the natural vector as well.
Lines 435-437. Activation is different from induction. Is there any other evidence in the literature of induction of RNAi-related genes during arbovirus infection?
Reviewer 2 Report
Comments and Suggestions for Authors
In this paper, de Mendonça et al assessed the infectivity and dissemination of a recently isolated Oropouche orthobunyavirus (OROV) strain in two widespread mosquito species, Aedes aegypti and Culex quinquefasciatus. Both mosquito species were refractory to oral infection, suggesting that natural transmission through these vectors is unlikely. Ae. aegypti showed viral replication and immune system activation, RNAi pathway was the most strongly activated in response to OROV infection in Ae. aegypti. In addition, they show that coinfection with CHIKV does not promote OROV oral infection. Considering the recent expansion of OROV circulation in South America, it is indeed crucial to study the vector competence of vectors circulating in this area. One of the main strength of this study is the use of field mosquitoes and an OROV viral strain isolated recently in Brazil. However, I have some concerns, mainly about the dose used to infect mosquitoes, which could represent a caveat explaining the lack of midgut infection and therefore competence. To make the conclusions supported by the data some controls should be performed. Or the caveats should be discussed. There are also a lack of references to support some statements.
Considering the mosquitoes were tested negative at 8 days post infection, it would have been beneficial to control for the initial ingestion of virus, ie sampling females minutes after the blood meal.
The OROV dose used to infect mosquitoes is quite low (10^5 pFU/mL). It is known that the dose given to mosquitoes is an important parameter for successful midgut infection, and that at least 10^6 to 10^7 may be required, at least in laboratory settings. The authors used DENV (and CHIKV) as a positive control, however, the dose used there is not described. Was the same infectious dose given for both OROV, DENV, and CHIKV? If the dose of DENV a,d CHIKV was higher, I d suggest the authors to repeat the DENV or CHIKV experiment using the exact same dose of virus to make sure that the dose is not explaining the lack of infectivity.
Figures 5 and 6: describe what is mock, ie mosquitoes not injected or mosquitoes injected with mock solution? If non injected mosquitoes, the injection alone could explain some difference in immune gene expression (due to healing or bacteria introduced during injection).
In the discussion, the authors suggest that OROV replication may occur in some certain conditions. Here, I suggest that the authors discuss the midgut infection barrier. It may not be certain conditions, but a midgut specific barrier, as seen previously with other viruses and mosquito combinations. Especially at the light of the low dose used to infect mosquitoes orally, the lack of infection following oral infection may be due toa dose dependent midgut infection barrier. If not trying higher doses of OROV, then this should be discussed in the discussion.
L435: no reference is supporting this statement. The authors shouodl be clearer as to what is consistent with other arboviruses, induced expression or antiviral role? In addition, induced expression does not mean a role in antiviral response. The authors should modify this section not to overstate.
Regarding Toll results interpretation, they show that the negative regulator is induced at later stages, therefore, Toll may be repressed at later stage and not activated. L447-448 should be modified accordingly.
L 454: “important questions regarding the potential for alternative transmission routes, such as venereal or vertical transmission” > How the mosquitoes could transmit vertically or venereally if they cannot acquire the virus orally in a first place?
Material and methods:
- OROV production is not described (CHIKV and DENV only)
- DENV titres are not described
- L275: Aedes mosquitoes> Aedes in italics
- L293: Previous studies conducted by our research group > reference to be added
- The dose used for each infection should be clearly described in legends or main text (for both oral feeding and injection)
- Gene expression in Aedes mosquitoes: describe at beginning of 3.4. that this was done following systemic infection.
Author Response
"Please see the attachment.

Round 2
Reviewer 2 Report
Comments and Suggestions for Authors
The authors addressed all my comments adequately.